# Food Habits and Forms of Food Insecurity among International University Students in Oslo: A Qualitative Study

**DOI:** 10.3390/ijerph20032694

**Published:** 2023-02-02

**Authors:** Charlotte Bauch, Liv Elin Torheim, Kari Almendingen, Marianne Molin, Laura Terragni

**Affiliations:** 1Faculty of Oecotrophology, University of Applied Sciences Fulda, 36037 Fulda, Germany; 2Department of Nursing and Health Promotion, Faculty of Health Sciences, Oslo Metropolitan University (OsloMet), 0130 Oslo, Norway; 3Faculty of Health Sciences, Kristiania University College, 0153 Oslo, Norway

**Keywords:** food insecurity, university students, international students, eating habits, health, food choices, commensality

## Abstract

A growing number of studies indicate that university students and especially international students are prone to experiencing food insecurity (FI). Still, few studies have investigated forms of FI among international students in Europe. Thus, this qualitative study aims to explore experiences regarding FI among international university students in Oslo. Sixteen semi-structured interviews were conducted between May and June 2022 and analyzed using a thematic approach. The sustainable livelihood approach (SLA) was used as a framework for analyzing and interpreting the data. The students experienced food prices as being high and found food variety at the grocery stores to be low, resulting in struggles to fulfil their food preferences and keep a varied diet. Particularly, social aspects of eating were affected due to high dining prices or inadequate cooking facilities in student homes. However, no student openly reported skipping meals and many mentioned attention for healthy eating. Considering our results, it seems of importance to give more attention to cultural and social aspects related to FI when assessing FI among international students. As the number of international students is increasing, knowing more about this phenomenon can support the promotion of initiatives addressing FI in this population.

## 1. Introduction

Food insecurity (FI) is defined as the “inadequate access to sufficient, safe and nutritious food to meet dietary needs and food preferences for an active and healthy life” [1]. The concept of food insecurity has progressively replaced the one of hunger, regarded as focusing too much on quantitative aspects of food scarcity and deprivation [2,3]. The Food and Agriculture Organization (FAO) identifies four main dimensions of food security: the physical availability of food, the economic and physical access to food, food utilization and the stability of the other three dimensions over time [1]. FI may therefore occur when the access to or availability of healthy, nutritious and culturally appropriate foods is compromised or when available foods cannot be utilized properly. These dimensions of FI enable a recognition of different contexts and forms in which food insecurity can occur [4] and an acknowledgement that there are various forms of FI that can be experienced in different life phases [5]. One of the most recent major increases in FI worldwide in various population groups occurred due to the global COVID-19 pandemic [6,7,8]. In Europe and North America, the prevalence of FI has increased for the first time since the beginning of the collection of data on FI in 2014 [6]. The number of people in continental Europe suffering from moderate or severe FI increased from 7.7% in 2019 to 8.8% in 2020 [6], indicating that access to safe food supplies and sanitation does not necessarily prevent people from experiencing FI [9]. Rising food prices, as have occurred during the COVID-19 pandemic, especially affect the food security status of people who spend a larger share of their income on food [10]. This includes university students, who, in general, have a low income. Reviews conducted between 2006 and 2021 show that, on average, 30–40% of university or college students in the United States of America (USA) were food insecure, compared to approximately 15% among the overall population [11,12,13]. At the beginning of the pandemic in 2020, approximately 20% of students in the USA reported a lower food security status than before [14,15]. International students (defined as those not born in the country they study in) were shown to be more likely to experience FI during the pandemic, as well as before it [16]. One review assessing the presence of related factors to FI, such as depression, stress or ethnicity [13]. However, none of the included studies considered lifestyle factors, such as the context of food choices, the food culture and food competence. 

To our knowledge, few studies have investigated FI and food behavior among university students in Europe, and none have conducted such research in Norway. As higher education in Norway is free from tuition fees, many international students choose to take a full degree or do an exchange study period there [17]. The city with the most students in Norway is Oslo, which accommodates approximately 70,000 students, including many international ones. Similar to other countries, Norway does not have any easily accessible support structures, such as study loans, for international students [18]. In addition, items such as food and beverages are more expensive in Norway compared to other European countries [17,19]. Therefore, international students may be at an especially high risk of experiencing FI. Despite the rising number of international students worldwide, limited studies have been conducted on their food security status [18,20,21,22,23]. Furthermore, most studies conducted to date are wide-ranging and quantitative and therefore cannot explore in detail the experiences of specific sub-groups, such as international students [24,25,26]. Reviews show that these studies often use methods such as the Food Security Survey Modules and the Food Insecurity Experience Scale (FIES) Survey Module [11,13]. These modules have formerly been criticized because they do not closely examine the dimensions of the availability or utilization of food, but focus mainly on one dimension of FI, which is the accessibility or lack of food [2,3]. Several studies have suggested the need to consider other aspects, such as cultural food identity or the social aspect of eating [3,27]. In addition, conducting more studies with a qualitative design has been suggested as these would enable an in depth exploration of factors related to FI, such as food choices, food culture and commensality [28].

One framework that may contribute to assessing these other FI dimensions, such as food choices and students’ culture, is the sustainable livelihood approach (SLA) [9]. It describes different capabilities, assets and activities required for a means of living and focuses on how households or individuals use six assets (human, physical, social, financial, natural and political assets) to ensure their livelihood, including food security [9]. Thus, the SLA may enable an examination of FI through a richer perspective. In fact, several studies have established a more holistic picture of FI by connecting it to the SLA and other concepts, such as vulnerability [9,29,30]. Using the SLA in connection with a qualitative study design has been shown to broaden the assessment of FI by enabling participants to address all issues connected to accessing, cooking and storing food [31]. Furthermore, through the SLA, contributing factors to, coping strategies for, and consequences of FI can be examined [9,32].

Coping strategies found among FI households are often connected to social capabilities and include having friends and family provide support in the form of meals [31].

The main aim of this study was to use a qualitative approach to investigate food choices among international university students in Norway and explore which forms of FI they may experience. Additionally, a connection between FI and SLA assets were explored. To examine this, typical food-related behaviors of international students, such as shopping and eating habits, were investigated, and how students cope with the new food environment was explored. Moreover, students were asked to describe differences between their food choices in Norway and their countries of origin (COOS) and how these differences may affect their food security situation while living in Oslo.

## 2. Materials and Methods

This study has a qualitative exploratory research design [33] and semi-structured interviews were conducted in Oslo. In a new field of studies where little work has been carried out and few definitive hypotheses exist, an exploratory approach is more suitable [34]. Theoretically, the research design is aligned with the overall purpose of qualitative research to achieve an understanding of how people make sense of and interpret their experiences [33,35]. Data were collected between the end of March and the end of May 2022.

### 2.1. Participants

Participants were recruited through purposeful sampling. Purposeful sampling is widely used in qualitative research for the identification and selection of information-rich cases for the most effective use of limited resources [34]. International students from the two public universities, University of Oslo (UiO) and OsloMet, completing their whole degrees in Oslo or studying abroad for a semester (i.e., Erasmus students) were eligible to participate in the study. We aimed at recruiting full-time students. Given the exploratory design of the study, our purpose was to recruit students with different nationalities, including low, middle and high-income COOs and different food cultures. Participants were recruited by posting an invitation on WhatsApp and Facebook, as these are quite common communication platforms for international students. In addition, colleagues, friends and participants were asked to recruit students from specific regions through snowball sampling [36]. Students were excluded if there have already been more than two interviews from the same country or geographical area. In total, 16 interviews were conducted.

### 2.2. Data Collection

A semi-structured interview guide was developed through a review of the existing literature on FI among university students during and before the COVID-19 pandemic by the first and last authors (CB and LT) of this paper. The literature research was conducted in PubMed and Jisc Library Hub using the PICO scheme (Section A.1). Afterwards, the interview guide was reviewed by the Norwegian Food Insecurity among European University Students during the COVID-19 Pandemic (FINESCOP) research team and other researchers from the university working on a similar project. The interview guide was pretested in three pilot interviews and revised according to the feedback obtained from these. During the revision, sub-questions were added, and the questions were restructured into different topics, based on recommendations for qualitative interviews [37]. The final interview guide consisted of six topics: personal information, one’s arrival in Norway, accommodation, financial resources and support, food/food consumption in Norway, the preparation of meals, the economy and food consumption, experiences of having to skip meals or not having money to buy food, and, finally, the impact of COVID-19 on food habits (Section A.2). The topic of food consumption in Norway was further divided into shopping habits and eating habits. Moreover, the interview guide included questions on the differences in students’ food choices and eating habits compared to their home countries. During the pretests, the last author (LT) was present, while the research interviews were held by the first author (CB). The pilot interviews were used as training for the first author (CB). After each interview, a postscript (Section A.3.) was filled in, where the interviewer recorded the most important findings and reflections. As no major changes were made to the interview guide, these pilot interviews were included in the final data set. To assure that the participants felt comfortable, the interviews were conducted either on the OsloMet campus or at other locations chosen by the interviewees. The planned duration of an interview was 45 min–1 h [38]. Each student was interviewed once and every interview except for one was conducted in English. The quotes from the non-English interview, which was held in German, were translated into English by the first author (CB), who is a native German speaker. The interviews lasted 52 min on average and were digitally recorded. Before starting the interviews, information was provided regarding the overall FINESCOP project and the qualitative sub-study, and informed consent was obtained from all participants. The data collection ended when the information given by students started to sound repetitive and new information produced little or no change to the developed codes indicating that data saturation was achieved [39].

### 2.3. Data Analysis

The collected interview records were transcribed verbatim with the help of the Transcribe App from the DENIVIP Group LLC and the MAXQDA coding program. The quality of every generated transcription was checked by using the original audio file. The interview transcriptions were anonymized by eliminating exact ages, study subjects and other personal information that could have made the students identifiable. The transcripts were then read thoroughly and coded using a thematic approach based on Clarke and Braun [40]. The software MAXQDA was used to help assign quotes to codes and to mark peculiarities. At first, three interviews were coded focusing on aspects related to FI and the SLA assets by the first author (CB) [40]. Afterwards, the evolved codes and themes were revised and discussed with the last author (LT), who co-coded parts of the interviews. Each interview was then coded using an inductive thematical analysis by the first author (CB) [40]. During this coding process, new codes emerged, and other initial codes were renamed or reassigned to other superordinate themes. After the preliminary coding of each interview, they were reviewed again to see if codes that arose from other interviews needed to be applied. Finally, the most relevant themes and subordinate codes were identified with respect to answering the research questions, and the data were screened again in relation to these codes and then analyzed in detail. The research questions were reappraised and adjusted where necessary. (Section A.4). 

The study was approved by the Norwegian Centre for Research Data (NSD) (number 821340).

## 3. Results

The 16 participants originated from 13 different countries: 6 came from non-European countries and 7 from Europe. Each country was represented by one participant, except for Pakistan and India, which were represented more than once. The participants were between 20 and 34 years old, and nine of them were aged 25 years or older. The participants with origins outside of Europe were mostly older than 25, whereas the European participants were mostly younger than 25 (Table 1). Seven of the students were female, and eight were male. While seven of the students had only come to Oslo for one semester, the other eight had stayed, or planned on staying, in Oslo for at least one year. Many students mentioned having purposefully chosen Norway as their study location because of the high standard of living and the well-known Scandinavian nature.

The main interview topics revolved around food habits while living as a student in Norway. Topics related to food security were introduced during the interview. When asking the students whether they were ever worried about having enough food or enough money for food, none of the interviewed students openly reported having experienced these feelings, nor did the students describe facing hunger or skipping meals.


*“I: But did you ever worry about not (…) having enough money to buy food? (…)*

*P: Uh, till now, I didn’t feel like that”*
(≥25, Ethiopia)


*“I: I’ve been worried about money in general, um, specifically, have I focused on food? Maybe not (…) I’ve never had to worry about food”*
(<25, United Kingdom)

Despite this, it emerged during the interviews that students faced challenges in terms of the availability of culturally appropriate food, the preparation of meals and the affordability of ready-made meals or restaurant foods. Furthermore, students experienced food prices as being high and expressed how they developed strategies for buying food on a budget. Finally, the students reported reducing the quality of their meals by narrowing down their diets to staple items and eating less varied foods.

These challenges and perceived differences in the food environment are described in more detail below, divided into experiences regarding shopping for food, preparing meals and eating and consuming foods in Norway (Section A.4).

### 3.1. Shopping for Food

#### 3.1.1. Higher Food Prices

After arriving in Norway, students tended to compare the price of food to the prices in their COOS to evaluate whether products were cheap or expensive. All the students, except for one (Interview 7 from Switzerland), described food prices in Norway as more expensive than in their COOS.


*“I noticed, yeah, straight away, like the local supermarket in the Sogn student village, it’s like a lot more expensive than what I’m used to (…) fruit and veg especially is (per item) (…) like one or two pounds more” *
(<25, United Kingdom)


*“I can’t find all the things (…) at a price that I find reasonable (…) I really love things like tofu and halloumi, but like, when I think of what I pay for it in Austria and what I pay for it here, (…) that’s just too much difference” *
(<25, Austria)

Students who were supported by their parents and described their budgets as flexible still chose to compare prices and not buy certain products they perceived as too expensive.


*“I know that I have my mom, and I can always text her—Hey, mom, I need some more money—but I don’t want to do that because I have my own savings, so I try to stick with my own budget” *
(<25, Croatia)

The students’ perceptions of cheap and expensive food seemed to change when they had jobs and got paid according to Norwegian standards.


*“The good thing is that I’m working now, so I can even afford to go out to eat” *
(≥25, Spain)

#### 3.1.2. Choosing Food Products Based on Low Prices

Price was one of the most often mentioned reasons for choosing which stores to go to and which products to buy. Students shopped at either low-cost supermarkets or ethnic shops located in an area of the city called Grønland. Many learned about Grønland from fellow students or at orientation meetings for the Erasmus network. Grønland was mostly visited for fruits and vegetables, while other food products, such as meat and dairy, were procured from the Rema and Kiwi supermarkets. Students were also willing to visit different supermarkets if the prices of certain products were lower.


*“So, like some things I buy at Kiwi, and some things I buy at Rema because of the price differences” *
(<25, Belgium)


*“So, uh, there are like, uh, if you buy like, uh, pulses and chickpeas and beans, (…) their prices are good. (…) Uh, their vegetables are (more) expensive (…) than the Grønland ones (…) don’t buy vegetables from there” *
(≥25, Pakistan)

#### 3.1.3. Balancing Price, Health and Convenience

Although the price was the main reason behind choosing certain food products, other reasons were also mentioned, such as product convenience, healthiness and quality.


*“I prefer to just buy (…) the mixed salads that, that are available at Kiwi (…). Convenience is (…) a very big part for me” *
(<25, Portugal)


*“I like Meny because the quality is better than, for example, Bunnpris” *
(≥25, Spain)

For example, students mentioned buying fruits and/or vegetables, even though they perceived them as being more expensive than in their COOS.


*“I also still wanna be healthy, (…) the fruits even though they are really expensive, I’m like yeah I still need them, so then I don’t really look at the price” *
(<25, Belgium)


*“Fruit is something like I would always, always buy” *
(≥25, Pakistan)

#### 3.1.4. Availability of Culturally Familiar Foods

Students mentioned struggling to find culturally familiar foods at regular grocery stores. Therefore, students, from outside of Europe in particular, mentioned going to ethnic stores in Grønland. However, while students from Pakistan reported being able to find all the products they desired in these stores, the student from Taiwan pointed out how, even in these special stores, common Taiwanese products were not available.


*“(I) actually can’t find things that I want in the Asian market (…) I went, but I still didn’t get anything because there is a really small selection” *
(<25, Taiwan)

Students from Europe also mentioned not being able to find all the products they would usually buy.


*“P: Um, yeah, cheese, definitely. I’m a huge cheese lover, but (…) I eat a lot less cheese here (…) because there are not that many options I would say” *
(<25, Croatia)


*“They (the supermarkets) are rather smaller. We have those huge supermarkets, but (…) I think the variety is the biggest, uh, difference” *
(<25, Croatia)

Furthermore, students from outside the EU perceived unfamiliarity in terms of the organization of grocery stores. Examples mentioned were not being used to buying meat and cheese in supermarkets or not being used to packaged and frozen foods.


*“There are supermarkets, but (…) we don´t go to these supermarkets (…) there is a huge market in Ethiopia, so in that market everything is available” *
(≥25, Ethiopia)


*“This was my first time I saw cheese here (…) to be honest, I’ve never seen, um, a general store having, uh, meat” *
(≥25, Pakistan)

Students changed their cooking habits and began preparing easy meals with staple items that they were familiar with and could buy cheaply in Oslo, such as rice and potatoes. 


*“Like if you have rice and beans, [that’s the] cheapest way. That’s a traditional Delhi dish” *
(≥25, India)

### 3.2. Preparation of Meals

#### 3.2.1. Preparing more Basic Meals

Many students described preparing more basic meals than in their COOS and perceived that the healthiness of their diets was negatively affected by this. In addition to less variety in terms of ingredients, students mentioned that, at home, their meals would be regularly prepared by family members who cooked a broader variety of food. Additionally, preparing food was cumbersome as the kitchens in student dorms or studios were not equipped with all the necessary utensils.


*“Back in Portugal, I’m used to such a (…) diverse raster of foods because (…) it would usually be my mother or my grandmother cooking, so they (…) have more time or more skills. So back (…) home I would (…) never eat like the same thing over and over [again], but here it’s a bit (…) harder, also because there are some ingredients that (…) I can’t find here” *
(<25, Portugal)

Some students reported that they had started cooking more often since they came to Norway. The motivation to cook varied, and while some students described themselves as food lovers that cooked every day, some reported how they only cooked because there was no other choice and they had never cooked before moving to Oslo. Two of the students told how they had been used to always eating out or ordering food in their COOS. However, since in Norway ordering food and going out is very expensive, these students had changed their habits and begun cooking meals at home.


*“And also, I don’t know how to cook at all, and I thought that I can just buy food easily here. But then things are all expensive and not so good [quality]. (…) It’s not just like slowly learning how to cook, but you have to immediately learn how to cook or else you have to eat outside every, every meal” *
(<25, Taiwan)

#### 3.2.2. Living Situations, Kitchen Facilities and Cooking Habits

Every student living in a dorm had a kitchen in their apartment with basic equipment, such as an oven, a stovetop with four hobs, a fridge and a freezer. Students described their kitchens as well equipped. Some were missing specific kitchenware, such as mixers, but the students stated that they did not see the point of buying them, as they were only in Oslo for a short amount of time.


*“I’m still trying to find like a mixer or something, but I don’t want to buy a new one because I’m leaving in three months. So, it wouldn’t really pay off” *
(<25, Austria)

The kitchen facilities influenced the motivation of students to cook. They did not want to spend much time cooking or, sometimes, could not cook because their shared kitchen was occupied by other people. Many students who lived in shared apartments stated that they did not like to cook at the same time as other people.


*“I do try and be quick just for, like, for my other housemates’ sake” *
(<25, United Kingdom)

Furthermore, the students’ kitchen facilities limited their socialization with friends, as they were not able to invite people over.


*“We don’t really have a common space where we can be. So, no table, no chairs where we can sit. (…) So I can’t invite people over. That’s really different to Vienna because I loved having friends over for dinner.” *
(<25, Austria)

Despite time and space limitations, most students made efforts to prepare meals that were healthy and talked about meals that included protein, carbohydrates and vegetables.


*“A combination of vegetables and, you know, pulses, rice, meat, everything in moderation” *
(≥25, Pakistan)


*“I try to have vegetables and protein and like rice or whatever” *
(<25, Croatia)

### 3.3. Eating and Consuming Foods

#### 3.3.1. Differences in Meal Structure

Some students reported cultural differences they perceived when considering their eating habits in Norway. These included eating cold meals for lunch and eating outside of the home regularly.


*“In terms of food, they (the Norwegians) prefer cold food for lunch. We never have cold food for lunch” *
(≥25, India)

#### 3.3.2. Missing Commensality

Participants reported that they felt like eating out or meeting with friends was not as common in the Norwegian culture as in their COOS. Students of European origin and students born outside of the EU likewise stated that they ate alone more often than in their COOS. Students from outside of Europe, in particular, reported being used to living with their families at home and therefore were not accustomed to eating alone.


*“People live together, so you share foods (…). It’s not common to eat from your own dish, (…) we eat from one dish together, especially at dinner times” *
(≥25, Ethiopia)


*“In Belgium, I would eat with my friends almost every day and, here, maybe once or twice a week” *
(<25, Belgium)

Most students said that going out to eat or drink with friends and family was an essential part of student life in their home country, which they had to cut back on here in Oslo due to high food prices. Another reason for not going out was the perceived low quality of the food served in restaurants. 


*“Here, probably, I go out once a week to eat and there [at home], I didn’t care. I just went out, I could even go out every day to eat. (…) because of the prices” *
(≥25, Spain)


*“In England, (…) going out for brunch, with friends (…) I did that at least once a week, (…) whereas (…) I wouldn’t even consider it here because all the places that look good (…) everything’s like 250 Kroners for the main meal” *
(<25, United Kingdom)

#### 3.3.3. “Forced” to Eat Healthier

A positive aspect of high food prices that was reported by students was that fast food or other unhealthy foods were not readily available on the streets. While this was common in the students’ home countries, they perceived that it was harder to access unhealthy fast foods in Norway as the prices were too high. Furthermore, several students stated that going out to have drinks with friends was more common and cheaper in their COOS. Due to this, some students described their eating habits in Norway as healthier compared to in their COOS.


*“It’s (unhealthy food is) available, but it’s not affordable” *
(≥25, Pakistan)


*“I: So would you say you drink more alcohol than at home?*

*P. Uh, no, less (…) it’s too expensive” *
(<25, Croatia)


*“We have a lot of those, um, like kiosks on the street where you get, uh, sausages and döner and all those things that are not good [for you] (…) here in Oslo, I have rarely seen any of those” *
(<25, Austria)

### 3.4. The COVID-19 Pandemic and Changes in Eating Habits

Most of the students had arrived in Norway during the COVID-19 Pandemic in 2021 or 2022. Therefore, they had not experienced the first major lockdown in Norway in March 2020. Only two students had been in Norway before the pandemic started. One of these students stated that the food security status of students had worsened during the pandemic and mentioned that she knew many students, Norwegians and internationals, that had experienced FI due to losing their jobs when the lockdown started in 2020. When asked about the impact of COVID-19, other students mentioned some general changes in their eating habits as a result of the pandemic. One of the most frequently reported change was moving back in with their parents. This change in their living situation led to changes in the students’ eating behaviors and meant that they ate more healthily, ate out less and cooked more often.


*“Eating much healthier, because [I did] eat out a lot before, and during COVID, I started making [food] at home”*
(≥25, India)


*“I moved back home then (…) I was cooking quite often, as there was not much to do while there was a lockdown” *
(≥25, Switzerland)

## 4. Discussion

In this qualitative study, we have explored how international students experience shopping for, preparing and consuming food in Oslo’s food environment, trying to understand aspects related to their food security. The sustainable livelihood approach (SLA) inspired the design and analysis of our data.

In contrast to the results of a former review, including studies conducted in the USA, none of the interviewed students openly reported experiencing severe forms of food insecurity, such as having to skip meals or going to bed hungry [13]. Nevertheless, students indicated that their eating habits were negatively affected by high prices and less variety at grocery stores, fewer opportunities for eating together and dining out, and cultural differences between the food environment in Norway and their home countries. Furthermore, students described consciously deciding not to buy all the foods they were accustomed to, thus limiting the variety in their diets. These findings indicate that students can experience challenges in fulfilling important dimensions of FI, such as meeting their food preferences, fulfilling their cultural food identities and the social aspects of eating.

These elements of FI are closely connected to a student’s livelihood assets (Table 2). 

The main findings are discussed referring to the SLA livelihood assets, with the exclusion of natural assets, as the natural asset focuses only on the safety of foods available.

### 4.1. Financial and Political Asset

The financial and political conditions have been seen at the core of the causes of FI. The literature investigating FI among international students indicates that financial constraints often affect international students more strongly than national ones [42,43]. Thus, financial constraints can have an impact on food choices and lead to poorer diets and students skipping meals [18,43]. An interesting finding of our study that warrants further investigation is how possible differences in income sources could have an impact on food security. In our study, for instance, Erasmus students were mostly supported by their parents, while students from outside of Europe were more likely to be financially independent from their parents. Previous studies indicated that students who are supported by their parents are less likely to experience FI [11,24]. However, the results of this study support previous findings, stating that even if parents are supporting their children while studying, the latter are sometimes reluctant to ask for more support [28]. As Norway is among the countries with the highest food prices [44], a non-Norwegian income and budget are likely to not be adequate with respect to Norwegian grocery prices and may increase the FI risk in students. Since the start of the 2022/23 academic year, international students are, under certain conditions, eligible to receive study loans [45]. None of the students interviewed for this study had the opportunity to apply for these loans. Furthermore, despite being eligible for such loans, many students have to work to be able to afford the Norwegian standard of living, and this is more often the case in Norway than in other countries in Europe [46]. The results from our study, although limited, seem to suggest that starting work and obtaining a Norwegian salary prevent students from experiencing FI, as they can widen their food choices and afford to eat out with friends more often. However, a consequence of working is that students have to cut back on daily chores, such as cooking food or meeting friends, as students often work at odd hours to fit in with their university schedules [24,47].

Previous studies and a systematic review found that FI was strongly associated with financial constraints and people choosing low-cost foods that would leave them with enough money to cover other expenses [11,24,28,48]. Our results suggest that this behavior is present among international students in Oslo but for a different reason. As the students were used to products being cheaper in their home countries, they found the prices unreasonably high and, thus, decided to abstain from some items during their stays. This phenomenon has been described as voluntary FI, in contrast to compelled FI [28]. Students described that they had room for prioritizing whether they buy food or spend money on other activities, such as clothes or going out. Therefore, they subjectively did not identify themselves as food insecure [28]. On the other hand, this behavior could be defined as food poverty or compelled FI, which describes a state where people have a low income and therefore are compelled to spend a large share of their money on food and limit other expenses [28,49]. Even students cutting back from some food items reported to spend the largest shares of their income on housing and food.

### 4.2. Physical Asset

Previous studies have shown a connection between students’ living situation, their income and their food security status [28,50]. The living situation of a student is a main part of the physical asset of the SLA [9]. In line with previous results, students reported that difficulties with sharing their kitchen or available facilities reduced their opportunities and motivation to cook and narrowed down their diet to basic items [18,28,51]. Not being motivated to cook could lead to students skipping meals or eating more snacks, which is a characteristic of students experiencing FI [18,28]. Furthermore, students reported that they often came home hungry, which could lead to food choices that favor foods with a lower dietary quality [52]. It should also be added that kitchen facilities are modelled for Norwegian students, which may not be suited for other food cultures. Previous studies among students from China and South Asia attending western universities indicated that the physical environment in kitchens represented a barrier to meal preparation [53,54].

### 4.3. Social Asset

Participants described their social contact with other students involving food (such as cooking, eating together or dining out) as limited. Students often talked about eating alone in their room, not inviting friends over for dinner as often as they did at home and limiting going out. The act of eating together and sharing meals, defined as commensality [55], is strongly connected to the cultural experience of being international residents [28,56]. Therefore, the findings of our study stand in contrast to the eating culture of students, which is characterized as having strong social relationships within their peer groups and often includes going out for food or drinks together [50]. Students from Africa and Southeast Asia, in particular, perceived that socializing while eating was not common, contrary to in their home countries, where it is an integral part of daily life [9,27]. Social contact with other students is not only an important part of student culture but can also decrease the FI rate, as students often ask their peer groups for help when struggling to afford food [24,26,57]. Furthermore, eating alone can lead to psychological problems and unhealthy diets [58,59].

### 4.4. Human Asset

The human asset of a student’s livelihood is closely connected to their food and nutritional knowledge and their motivation and interest in food [9,29]. A previous study indicated that students who are food secure are more likely to feel confident about being able to cook a variety of healthy meals and switch out ingredients to make meals more healthy than food insecure students [60]. Several students interviewed for this study reported not knowing how to incorporate products available in Norway into their diets. Furthermore, students struggled to cope with missing ingredients or kitchen facilities and, therefore, failed to fulfil their food preferences. This could be a barrier to food security since this can have an impact on diet quality. When asked what foods they bought at the supermarket, many students described basing their choice on the healthiness of products. A study conducted among international students in Belgium shows similar results, as students report eating many fruits, vegetables and whole grain products [61]. This is an uncommon finding, as usually university students tend to rely more upon fast food and ready-made meals when moving to university or away from home to study [62,63]. Moreover most prior studies found that students refrained from buying healthy foods, such as fruits and vegetables [28,64,65]. Further studies should evaluate this phenomenon.

Another aspect closely related to the human asset is eating culture. Students reported that they had problems preserving their usual eating habits, including not only which food items are bought, but also how they are prepared and how and when they are eaten. People of non-European origin, in particular, reported not being used to buying their food through supermarkets where food is packaged in plastic or frozen [27]. Furthermore, students reported not being able to find all the products they would usually buy at a Norwegian grocery store and perceived the general product variety as low. These challenges can be particularly relevant in Norway as the Norwegian retailers’ system is characterized by high store concentration and low product variety [19]. 

Furthermore, students from Africa and Asia reported not being used to eating cold foods, such as sandwiches, as their main meal, which is the typical Norwegian lunch [66]. In particular, the students from these countries reported eating all meals at home. One reason for this might have been that they could not identify themselves with the food offered at the canteen. As this study only included a small number of students from Africa, South Asia and the Middle East, aspects related to the origin of students need to be more closely examined in further studies. In general, the differences in the food environment experienced by international students were similar to experiences reported by migrants or other ethnic minorities living in Oslo [27]. These experiences have been previously described as cultural FI in a qualitative study on food insecurity among minority college student groups [67].

### 4.5. Strengths and Limitations

This study provides a first overview of the experiences of international students, yet it does not represent the composition of international students in Norway. Whereas South Asian students were numerically well represented, East Asian students were underrepresented in this study, based on the knowledge that approximately 30% of the international students in Norway come from Asia [68]. In addition, we were not able to recruit students from Central or North America. Furthermore, most of the countries included were only represented once. It would have been desirable to recruit more participants from each country and especially more students with origins outside of Europe. To recruit more students, different recruitment methods could have been used, such as visiting language classes or student dorms. As a limited timeframe was available to conduct the interviews for this study, the opportunities to recruit people from different locations were limited. However, by using purposive and snowball sampling, it was possible to recruit students very specifically to ensure the inclusion of as many countries and continents as possible.

Another limitation of this study was that the analysis of the interview data was mostly undertaken by one researcher (CB); however, another researcher (LT) participated in the coding of the first interviews and supervised the analysis process. 

A strength of the study was that the interviewer (CB) was herself an international student in Norway. Therefore, she was able to create a good connection with the participants and they were willing to open up during the interviews. Nevertheless, there were some aspects that could have led to bias. Firstly, the researcher conducting most of the interviews (CB) had limited training in interviewing. Secondly, students knew that the researcher had a background in nutrition and therefore a desirability bias may have been present. 

Another limitation of this study is that the recruitment was mainly based on the country of origin and differences between the kind of degree (bachelor or master) or the kind of enrolment (as Erasmus or full degree student) were not included in the recruitment criteria. As these may have a potential impact on experiences with FI, more efforts to enlarge the sample could have been made. This was, however, not possible because of time constraints.

Qualitative studies on FI have so far mostly been conducted in North America and Australia. Despite its limitations, this study provides important and unique insights into the topic of FI among international university students in Europe and introduces important themes to be considered in further research.

## 5. Conclusions

International students are a group potentially vulnerable to FI. Food insecurity includes different dimensions and while quantitative aspects of FI have been widely investigated, less is known about qualitative aspects, such as fulfilling cultural preferences. The adoption of the SLA in this study allowed it to move beyond previous ways of assessing FI and showed that linking FI to the financial, physical, human and social assets of international students’ livelihoods can contribute to a wider understanding of forms of FI. Our findings indicate that international students experience forms of FI connected to socio-cultural aspects. As many international students struggled with preparing nutritious, culturally familiar and affordable meals in company with others, initiatives aimed at promoting familiarity with the new foods and food environment should be implemented. Additionally, students’ socialization with others mediated through dining together could be improved. As the study seems to suggest, there are differences between the experience of Erasmus students and students enrolled for a whole program, and between students coming from Europe and students coming from non-European countries. Further studies considering these groups separately are recommended. 

## Figures and Tables

**Table 1 ijerph-20-02694-t001:** Participants in the study.

Interview	Country of Origin	Age	Studies in Oslo
**1**	Austria	<25	Erasmus Exchange Student (Bachelor)
**2**	Belgium	<25	Erasmus Exchange Student (Master)
**3**	Ethiopia	≥25	Full-Time Master Student
**4**	Pakistan	≥25	Language Course Only
**5**	United Kingdom	<25	Erasmus Exchange Student (Bachelor)
**6**	Portugal	<25	Erasmus Exchange Student (Bachelor)
**7**	Switzerland	≥25	Erasmus Exchange Student (Master)
**8**	Pakistan	≥25	Fulltime Master Student (Joint Master)
**9**	Croatia	<25	Erasmus Exchange Student (Bachelor)
**10**	Spain	≥25	Full-Time Master Student
**11**	Taiwan	<25	Exchange Without Stipend (Bachelor)
**12**	India	≥25	Full-Time Master Student
**13**	Peru	≥25	Full-Time Master Student
**14**	Kenya	≥25	Full-Time Master Student
**15**	India	≥25	Full-Time Master Student
**16**	Pakistan	≥25	Full-Time Master Student

**Table 2 ijerph-20-02694-t002:** Relation between assets of the SLA and aspects affecting FI among international students.

Assets of the SLA [41]	Aspects Related to FI Relevant among International Students in Oslo
Financial Asset	Higher food prices, source of income, kitchen facilities
Political Asset	Source of income ≥ shopping more freely with a Norwegian income
Physical Asset	Kitchen facilities
Social Asset	Eating culture, commensality, timeframe for cooking, higher food prices
Human Asset	Eating culture, food literacy, wanting to eat healthily

## Data Availability

The data that support the findings of this study are available on request from the corresponding author (L.T.).

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
