# Peer review of "Food Habits and Forms of Food Insecurity among International University Students in Oslo: A Qualitative Study"

_ijerph, 2023, doi:10.3390/ijerph20032694_

Round 1

Reviewer 1 Report

There are some areas where use of English is incorrect. For example, in the Abstract, the authors use "majorly". This is not typically used in that manner and should be edited. Also, "fivteen" is not an English word. Perhaps have someone fluent in English read over the manuscript from a technical perspective. "Behavior" is spelled with European spelling which may or may not be an issue with the journal. Just noting this. (Most of the paper is fairly well written. I don't mean the authors don't speak English well, there are simply some technical misuses.)

Some abbreviations are not spelled out so readers will not know what these mean. Examples: FINESCOP- line 99, OsloMet- line 152, MAXQDA--line 137-38. In addition, the first reference to what "EU" means in on line 173, and it is stated to mean "Europe" but Europe is noted much earlier in the manuscript so it needs to be abbreviated earlier.

Software, or applications probably need to be referenced at least with manufacturer and city of the parent company and version in parenthesis. Example: TranscribeAP (Company name, version, city) , MAXQDA - "MAXQDA (VERBI Software, Version X, Berlin)".

In the Methods section, it is noted that 15 interviews were done - line 103. However, by line 154, it states that "16 participants" were invovled. Which is it? By line 538 we are back to 15. This must be clarified.

Line 509 starts a new paragraph but reads as though the thought is a continuation of previous commentary. This should be edited.

Line 523 has a 'methods' sub-heading. That seems out of place here. I looks like Discussion and may need a sub-header but it does not seem to be related to method.

References--some of the journal names are spelled out completely, and some seem to use AMA abbreviations. They need to be edited to conform to this journal's specifications. Also, the 48th citation in the reference list has the title in all caps, which is not appropriate.

Author Response

Thank you for very useful comment. here you find a detailed point to point answer!

Reviewer 2 Report

Thank you for the chance to review your study. It is exciting to see this qualitative study, especially as part of a master's thesis. I believe that this research study is important, and I have provided feedback to strengthen the reporting of this qualitative study. Unfortunately, the experiences of international students have been overlooked in research studies, even in the U.S., where many studies on food insecurity and college students have been completed. I hope you will consider my feedback, along with the other reviewers, and resubmit a revised manuscript.

Overall

·       This manuscript reports on a qualitative student from a student’s masters-level thesis project. Given a tight timeline, with all data collection completed in two months (end of March to end of May 2022), and the summer and early fall for analysis, there may have not been enough time to fully develop this manuscript. While the topic and research aims/questions are important, this manuscript requires major revisions to be ready for publication.

·       The aim doesn’t fit with the instruments and presentation of results.

·       It’s not clear how the SLA influenced the study design and data collection, including design of instruments or codebook for analysis, or how the SLA influenced data analysis.

·       There are major issues with methods and results and the introduction and discussion sections need to be rewritten.

·       There are places that need conceptual clarity – hunger vs. food insecurity vs. food security and source of definition, if keeping focus on food insecurity, which I don’t recommend, because manuscript isn’t focused on food insecurity.

·       Please see comments below.

introduction

·       Please rewrite to ensure framing is assets/strengths-based and fits with study design and approach to data collection. In my opinion, the focus on food insecurity doesn’t fit with content presented in methods and results, especially the description of food insecurity measurement (lines 65-89).

·       Please consider framing this manuscript as an application of the sustainable livelihood approach (SLA) to understand the food choices of international students, and their community, university, and home food environments as international students after beginning studies in Oslo, a large urban city, in Norway.

·       When writing the introduction (and discussion), please ensure that you’re citing the original research papers and not relying on review papers or loosely related articles. For example, when describing food insecurity measurement, the citations are not the ones I would have expected for the US FSSM and the FIES.

methods

·       Please explain how “sustainable livelihood approach (SLA)” was used to develop interview guide and other instruments for data collection and used in data analysis and interpretation. The SLA was was mentioned in introduction and opening of discussion section.

·       Study design is missing. Please describe study design as in-depth interview study.

·       Some text in this section doesn’t go here, but goes in results. For example, lines 105-107 about “Seven of the students had only come to Oslo for one semester…” goes in results.

·       Sample and recruitment – insufficient detail and justification

o   I don’t understand why there wasn’t more detail for sample and recruitment if it was important to achieve a diverse sample of students from different continents. It seems more important to consider whether country of origin was a high, middle, or low income country and other characteristics including food insecurity status currently.

o   Please describe why a convenience and snowball sampling strategy was used. In general, this is not recommended for qualitative research. I understand if this sampling strategy was based on practical constraints (e.g., limited time for a thesis project), but the strategy needs to be explained.

o   Please explain why only students at two universities were eligible for inclusion and list.

o   Lines 110-111 describes the aim to recruit students of different nationalities and students from different continents. Please justify this focus and explain inclusion/exclusion criteria and recruitment strategies to achieve this aim. It’s not clear.

o   It’s not clear if study focused on undergraduate and graduate students, or only on graduate/post-graduate students. Please justify with citations and explain.

o   It’s not clear if study focused on part-time vs. full-time students. Please justify with citations and explain.

o   From results, nearly all participants spoke English and completed the interviews in English. What were the language requirements for inclusion or exclusion criteria?

o   Please ensure that all inclusion and exclusion criteria are explained with citations as needed to justify decisions.

o   How were prospective students/participants screened to determine if they were eligible? Please detail the screening process. It’s not clear.

o   Were participants provided with an incentive? If not, why?

·       Survey/questionnaire

o   The sociodemographic survey/questionnaire wasn’t described in the methods, though characteristics were shown in Table 1 and summarized (as expected) in opening paragraph of results. Please provide a complete report of data collection including screening calls, and all instruments used to collect data. It’s important to also justify the questions included in the sociodemographic survey/questionnaire.

o   Was food insecurity measured with FSSM or another tool? If not please justify not measuring food insecurity quantitatively and using data for interpretation?

o   Were any questions asked about individual, household/roommate, or community/university, or government support or assistance programs? It’s not clear how many students, if any, had stipends for living expenses, jobs, etc.

o   Were any questions asked about participants’ family responsibilities and if participants moved to Norway with a partner, children, or other family members? In food insecurity studies, it’s common to see a description of household composition, but this wasn’t described. It’s well known that students may have unique living situations.

o   Were any questions asked about students’ lives before? Were they from smaller towns and more rural communities vs. large metropolitan urban areas?

·       Interview guide

o   Please provide references to theoretical or conceptual models and studies used to inform the interview guide, including the senior investigator’s prior studies. There were no references provided in description of interview guide. It’s critical that the reader understand the theoretical and empirical foundation of the interview guide.

o   Thank you for including the interview guide as a supplementary file. Please reference Supplementary Info for interview guide (described on lines 122-124).

o   Some of major questions appeared to be closed-ended and not open-ended questions (recommended for qualitative studies). Please explain how questions enabled participants to share their experiences.

o   The interview guide looks lengthy for interviews between 30 minutes and 1 hour and 20 minutes. It’s important to give enough information to show that interviews had richly detailed qualitative data for this analysis.

·       Interviews

o   Please describe the person or the team of people who did the interviews. Describe their race/ethnicity or immigration experiences, age, language(s) spoken, or any other relevant characteristics related to subjectivity. It’s not clear how if interviewer(s) would have been considered part of the inside group of interest (international students studying in Norway) or an outsider (not an international student studying in Norway).

o   Also, please include any training that the interviewer(s) had prior to conducting interviews and techniques used for creating rapport, probing, being non-judgmental etc.

o   Please provide more details on the interviews.

§  Please state that all participants were interviewed once. What was the target duration of an interview? Typically, the goal for an in-depth interview is 1 hour – 1.5 hour.

§  Please report average duration of interview. A 30-minute interview is very short, especially considering such a lengthy interview guide.

o   Were interviews conducted remotely or physically in person? It seems they were physically in person.

o   Please update description of recording. I’m sure recording was done with a digital recorder and not a “tape recorder.” (see line 131)

o   Were fieldnotes prepared to document observations and reflections from the interviewer? This is very important for interpreting findings from analysis. I didn’t see any description of fieldnotes.

o   Saturation

§  Lines 133-135: Important to talk about saturation in a qualitative study, but text says students from every non-European continent with low-to-middle income countries were included. There were no students included from low and middle countries in Central America (1 student from South America). In addition, there was only one student included from Africa. Of the 15 or 16 students, 7 were from EU countries.

§  With a long interview guide and relatively short interviews and no details on target duration or average duration of interviews, it is not clear that saturation was achieved. Please talk about limitations of doing analysis with one person.

·       Saturation

o   Lines 133-135: Important to talk about saturation. XXXXX

·       Transcription and translation

o   High-quality transcription and translation are key.

o   Were verbatim transcripts created and used for analysis? It’s not clear.

o   Please provide more justification for Transcribe app for generating a verbatim transcript for research purposes and how the MaxQDA program was used for transcription. Include citations for the app/software.

o   Were audio files used to check quality of transcription? Please provide details.

o   I’m concerned about the description of the transcription for the German language interview. Text says that the “quotes were translated into English.” Who translated the one German interview? Was a complete transcript created? Verbatim transcription? How was the translation checked? Please provide details.

·       Analysis

o   This section needs more development. The description of analysis is inadequate. It’s possible that additional analyses are needed for this manuscript.

o   This section must describe process for creating a codebook, that is based on the literature and investigator’s prior experiences, and outline creation of a priori and emergent codes. The section on coding needs more development with citations to literature as needed.

o   In addition, this section needs to include memo-writing and documentation of initial observations, insights, etc. from reading, discussion, and coding of the transcripts with citations to literature as needed.

o   Lines 141-142: “three interviews coded inductively.” Please explain what this means. Open coding? No a priori codes were used, only emergent or in vivo codes were created? Include citations to coding techniques.

o   Please provide more detail on number of coders and extent of co-coding, if any. It is expected that either all interviewers were double coded (two coders) or there was co-coding (one main coder did all coding and another coder coded portions of transcripts to ensure accuracy and reliability in application of codes.) If transcripts were only coded by one coder, then this raises serious concerns about potential of bias based on one coder’s interpretation and mistakes in coding (e.g., not applying a code when needed, applying a code inconsistently.)

o   What strategies were used for data interpretation? How were quantitative data used in this process? Fieldnotes? Any peer debriefing or member checking? Please describe and explain.

results

·       Please ensure that results are presented consistently with study design (e.g., grounded theory or phenomenology study vs. an in-depth interview study) and aligned with interview guide and other instruments. Based on my understanding, this study was not about food insecurity but about the food choice process. 

o   After re-reading results, this section presents more findings related to the food choice process and not experiences of food insecurity. It may be helpful to reframe using theory on the Food Choice Process Model (by Sobal, Bisogni et al. from Cornell University) and present findings organized by planning, shopping, cooking/preparation, eating, and post-eating activities like cleaning/storing food, etc. and comparison of food choices between current and former food environments (both home and community environments).

o   The interview guide didn’t focus on food insecurity, but on parts of the food choice process specific to the community, campus, and home environments of international students. In addition, with almost no information given for sampling frame and inclusion/exclusion criteria, it’s not clear that students included students who may have dealt with food insecurity before their studies in Norway or currently as international students in Norway. Given that, it doesn’t make sense to have statements saying that students didn’t talk about food insecurity or describe not having enough to eat.

o   The second and third paragraphs of results are not findings. Parts are very general and restate the topic of the study. There is not support for statements.

·       Overall, there needs to be more support for statements with consistent description of terms like some, most, many. For example, what does “most” mean here? 12 out of 15/16? It’s not clear. Please consider adding a Supplementary Table/Figure that provides additional exemplar quotes for major/minor themes to provide evidence for findings.

·       Please consider reframing results in a way that is not Euro-centric given focus on international students and more person-centered.

o   It’s not clear why students were broken up into European and non-European international students. Please update introduction and methods to provide more justification.

o   It’s not clear why Table 1 and short descriptions of each participant (e.g., [<25, European (EU)]) used a cutoff of 25 years. Please consider providing the actual age in the short descriptions. If insist on keeping the 25 yr cutoff, then justify decision with support from literature.  

o   For briefly characterizing quotes from people, then I suggest something like this: [participant 1/pseudonym, male/female student, age, country of origin]. Please consider including their actual age, gender, and country of origin. If you believe it’s critical to show if they are EU vs. non-EU, then include that at the end of the list like this: [participant 1/pseudonym, male/female student, age, country of origin, EU/non-EU].  

o   For example, please consider assigning a pseudonym for each participant with a common name from their country of origin. If this doesn’t feel “right” then assign them an anonymous study identifier like “student 1”.

·       The sociodemographic survey/questionnaire wasn’t described in the methods, though characteristics were summarized (as expected) in opening paragraph of results. Please make sure that this is added to methods.

·       Was food insecurity status measured with FSSM or another food insecurity tool? It’s not clear. If it was measured, then please report this. If it wasn’t measured, then please justify in methods.

·       Please add a couple of sentences in first paragraph of results to summarize the nature of studies in Norway and describe how the situations may be different based on different studies: 8 fulltime master student (including 1 join master), 6 erasmus exchange students, and other studies (e.g., 1 language course only, 1 exchange without stipend.) Please ensure that description refers to whether students were part-time or full-time, or whether students were graduate/post-graduate students vs. undergraduate students. Without more detail about the included participants, it’s hard to interpret the findings and think about how they compare to findings from other studies (on food environment or food insecurity and college/university students) from the U.S. or Australia.

·       Table 1 doesn’t add value for a qualitative study like this. Please consider adding as a supplementary file if requested by other reviewers, or if it’s personally important to include this table.

o   There were 16 participants listed in the table, but the abstract and in discussion (see section 4.5 limitations line 538), the manuscript says that there were 15 participants. Please clarify the sample size.

·       Table 2 doesn’t add value for a qualitative study like this. The content looks like an excerpt of the codebook, where a priori and emergent codes were aligned with major activities that are part of the food choice model. However, it is critical to include a description of codebook development and provide more details for analysis, including coding.

·       I am not providing additional comments for results section because it needs to be rewritten with more support and potentially, additional analyses.

DIscussion

·       In general for discussion, this section needs more support from literature. The references list looks comprehensive and relevant to topic and research question, but this section needed more time for development.

·       After the first read, I was confused by presentation of discussion.

o   With the Figure 1, this is more common for a phenomenology study or a grounded theory study, but I was not expecting this for a less intensive qualitative study design (in-depth interview study with 15 or 16 participants with one 30 minute-1 hour interview). If want to include a figure like this, then there needs to be more description of analysis/results to support it.

o   Some parts that detail the figure read more like results. Please consider reorganization so it’s clear which portions are results and discussion.

o   Was the figure based on SLA? If so, need to clearly define how SLA was used to frame the study, conduct data collection and analysis, and guide presentation of results.

o   Is Figure 1 supposed to a model of SLA specific to international students? If so, then it needs much more support/development from results and conceptual clarity.

·       The first paragraph mentions the “sustainable livelihood approach (SLA)”, but that approach was not defined or supported with citations. It would have been better to describe how this SLA approach was used to guide the study at the start of the methods section, along with a justification for why this approach made sense.

·       There is a need to conceptually distinguish between food insecurity (as a continuum of experiences) and hunger and be consistent in language. See line 411 for reference to hunger.

·       Line 417, there is mention of “other studies” but a reference to ref #8 (a review paper). Normally, the citations for the original research would be cited and not a review paper. If prefer to cite a review paper, then need to rewrite sentence so it’s clear. In addition, given nature of this study with focus on international students at a univeristy in Europe, it’s important to describe the samples/settings of the studies mentioned, so reader can follow along. (Ref #8 looks to be a review of U.S.-based studies with focus on student athletes.)

·       One concern with results is that the findings reflect a sample of students who had greater access to resources and opportunities and were able to move to Norway and adjust with more ease. Without more information about students’ socioeconomic status or lived experiences or other contextual factors, it’s harder to make sense of findings. What are common characteristics of international students studying in Norway during a peri-pandemic period? How does we/readers know that students represented a typical international student studying in Norway?

·       Line 523 – section called 4.4 Methods. I’m not sure about the purpose of this section. Is it implications for research, practice, and policy? Parts of it read more like limitations and has little support from the literature. Please rewrite discussion to better organize and develop main points presented here.

·       Line 537 – section called 4.5 Limitations. This section needs more development. It reads like a first draft. There needs to be more citations to connect with literature. In addition, there needs to a discussion of threats to validity and reliability. The last paragraph in this section seems to transition to a strengths section, which doesn’t exist.

·       There is no clearly defined section for Strengths, which is a missed opportunity. Please add a new subsection for strengths and bring in relevant studies as needed.

conclusions

·       Please review and review after making changes to other sections.

References

·       Relevant sources cited.

·       High-quality journals in public health nutrition

·       Experts on food insecurity and college students including M. Burening, CJ Nikolaus

·       Please revise references to address all issues noted below.

o   Please check for duplicate references (e.g., ref #6 Bruening is repeated as ref #32) and remove them. Update in-text citations and references as needed.

o   There are many formatting errors with references. Several references have errors with author/publisher names, sentence case of article titles, or are missing information for journal name, volume number, or DOI. Please ensure that the revised manuscript includes complete reference information for all sources cited. I’ve provided a list of references to check for convenience: ref #8 (no vol #), #9 (no abbreviated journal name), #11 (incorrect journal abbreviation), #13, 15, 18-22, 24, 25, 27-30, 32 (duplicate w/ref #6), #39, 44-46, #48, 49 (define abbreviation for publisher name and ensure that all details are provided.)

o   Please check journal’s formatting rules for websites and online reports. Some references do not appear to provide complete information for source (e.g., ref #1-3, #13, 15). Please make sure to define acronyms, which is very important for international audience and ensure that author, publisher or organization, and any relevant details are included, especially URL and dates updated and accessed.

o   Please check journal quality for journal named “Jonus” in ref #48. I checked Scimago – JR website, but I couldn’t find the journal. https://www.scimagojr.com/journalsearch.php?q=Journal+of+Nusantara+Studies+

      SUPPLEMENTARY INFORMATION

1. Thank you for including the interview guide.

2. Please add a completed COREQ checklist as a supplementary file to show quality of reporting of qualitative study and reference checklist as Supplementary Table/Figure in methods section. COREQ (Consolidated Criteria for Reporting Qualitative Studies) is a checklist used by Journal of the Academy of Nutrition and Dietetics. If there is another EU-focused checklist for qualitative studies, then that's fine. I believe this would help in the reporting. https://cdn.elsevier.com/promis_misc/ISSM_COREQ_Checklist.pdf

Author Response

Dear reviewer, we appreciated the work you have done in commenting our article. You find attached a point to point answer. The revised article, with track changes has taken your comments in great consideration and we think that they contributed to an overall improvement 
